# A landslide susceptibility map based on spatial scale segmentation: A case study at Zigui-Badong in the Three Gorges Reservoir Area, China

**Xianyu Yu**[ID]*, **Huachen Gao**

School of Civil Engineering, Architecture and Environment, Hubei University of Technology, Wuhan, Hubei Province, P.R. China

* yuxianyu@hbut.edu.cn

**Data Availability Statement:** The original data of all the map data used in this paper are confidential in China, including the 1: 10,000-scale landslide hazard map, 1: 50,000-scale topographic map, and

## Abstract

China experiences frequent landslides, and therefore there is a need for landslide susceptibility maps (LSMs) to effectively analyze and predict regional landslides. However, the traditional methods of producing an LSM are unable to account for different spatial scales, resulting in spatial imbalances. In this study, Zigui-Badong in the Three Gorges Reservoir Area was used as a case study, and data was obtained from remote sensing images, digital elevation model, geological and topographic maps, and landslide surveys. A geographic weighted regression (GWR) was applied to segment the study area into different spatial scales, with three basic principles followed when the GWR model was applied for this propose. As a result, 58 environmental factors were extracted, and 18 factors were selected as LSM factors. Three of the most important factors (channel network basic level, elevation, and distance to river) were used as segmentation factors to segment the study area into 18 prediction regions. The particle swarm optimization (PSO) algorithm was used to optimize the parameters of a support vector machine (SVM) model for each prediction region. All of the prediction regions were merged to construct a GWR-PSO-SVM coupled model and finally, an LSM of the study area was produced. To verify the effectiveness of the proposed method, the outcomes of the GWR-PSO-SVM coupled model and the PSO-SVM coupled model were compared using three evaluation methods: specific category accuracy analysis, overall prediction accuracy analysis, and area under the curve analysis. The results for the GWR-PSO-SVM coupled model for these three evaluation methods were 85.75%, 87.86%, and 0.965, respectively, while the results for the traditional PSO-SVM coupled model were 68.35%, 84.44%, and 0.944, respectively. The method proposed in this study based on a spatial scale segmentation therefore acquired good results.

## Introduction

Located on the eastern edge of the Asian continent, China, with active geological tectonic movements and a complex geological environment, is a country that experiences frequent

1: 50,000-scale geological map, we have no way to obtain them online in publicly. However, after declassification processing, these map data used only in this study area can be obtained by contacting the corresponding author and Youjian Hu, professor and doctoral tutor at the School of Geography and Information Engineering, China University of Geosciences (Wuhan) (imxg@cug. edu.cn). All other data (landsat-8 data, ASTER GDEM data, seismic activity data and atmospheric rainfall data) can be downloaded through the download link in the paper.

**Funding:** Funded by X.Y. 41807297; Q20171410. National Natural Science Foundation of China; Hubei Provincial Department of Education. http://www.nsfc.gov.cn/; http://jyt.hubei.gov.cn/. The funders had no role in study design, data collection and analysis, decision to publish, or preparation of the manuscript.

**Competing interests:** The authors have declared that no competing interests exist.

geological disasters such as landslides. During the 15 years from 2002 to 2016, 275,787 landslides occurred in China, accounting for 73.0% of all geological disasters in the country. In 2018 alone, there were 2,966 geological disasters around the country, including 1,631 landslides, i.e., 55.0% of all geological disasters that year.

A landslide susceptibility map (LSM), as a non-deterministic method of prediction, is currently the main method used for the prediction of regional landslides. Using the engineering geological analogy method, an LSM can be obtained through the use of a mathematical model to determine and assign the degree of importance of the LSM factors that cause landslides. Mondal and Mandal used a logistic regression (LR) model to evaluate landslide susceptibility in the Balason River Basin in the Indian Darjeeling region of the Himalayas. The result showed that the LR model can be used for landslide hazard research and decision making [1]. Wang et al. compared several methods for constructing an LSM, such as the frequency ratio (FR), LR, decision tree (DT), weights of evidence (WE), and artificial neural network (ANN) in a study in Mizunami, Japan, and found that LR had the best area under the curve (AUC) value [2]. Aditian, Kubota and Shinohara used three methods, FR, LR, and ANN, in a study of landslides triggered by heavy rains in the Ambon region of Indonesia: the study showed that the ANN had the best results among these three methods, and was the best method for interpreting the relationship between landslide and LSM factors [3]. Saro et al. used two methods, LR and ANN, for the construction of an LSM in Inje City, South Korea, with the results indicating that the accuracy of the ANN was higher than that of the LR model [4]. Hong et al. compared the effects of four support vector machine (SVM) models based on different kernel functions in the LSM by taking Luxi City, Jiangxi Province, as a study area. The results showed that the SVM models using these four different kernel functions achieved good results, with the model using a radial basis function (RBF) as the kernel function having the best effect, regardless of the success rate or prediction rate [5]. Pham et al. conducted an LSM study in Pauri Garhwal, India, and compared the SVM model with four Bayesian algorithms: the naive Bayes tree, Bayes network, naive Bayes, and decision table naive Bayes models. The analysis results showed that the SVM model had the best predictive performance [6]. Despite having achieved acceptable results in their application, such methods tend to ignore the spatial distribution of landslide hazards and extends them to the entire study area without considering the spatial applicability of the models. This affects the selection and assignment of important evaluation factors, and thus reduces the accuracy of the LSM.

To overcome the above problems, LSM methods that consider the spatial scale of landslides have emerged. About 20 years ago, Fell et al. published LSM guidelines. The authors believed that landslides of different scales should be evaluated at the corresponding spatial scale, and that the selection of the LSM factors should have a scale that is compatible with the spatial scale [7]. In the same year, Cascini affirmed the guidelines proposed by Fell et al. and focused on the applicability of the susceptibility and hazard zoning of landslides at different scales. In this study, according to the scales and applications of landslide zoning, landslides were divided into two categories: small & medium scales and large & detailed scales, and the results indicated that the guidelines were a "powerful tool for landslide and hazard zoning at different scales" [8]. Paudel, Oguchi and Hayakawa extracted the best scale of each LSM factor using the random forest model, and then constructed an LSM. Their experimental results for Niigata and Ehime prefectures in Japan showed that a multi-scale LSM model was superior to a traditional model [9]. Schlögel et al. extracted LSM factors using a digital elevation model (DEM) with different precisions (5, 10, and 25 m), and the experiments found that the LSM factors for the DEM with 10-m precision was the best data combination for acquiring the LSM [10]. These methods explore the relationship between the spatial scale of the LSM and the selected data accuracy, sampling accuracy, and applicable range, and promotes research on the spatial

scale of LSMs. However, these methods weaken the concept of spatial scale in the scale or resolution of an LSM, and do not analyze the differences between the LSM at different spatial scales or consider the essential importance of such differences. They also ignore the definitiveness of spatial scale to the production of an LSM.

Some researchers have used the geographic weighted regression (GWR) model to overcome these problems. Zhang et al. employed the GWR model, and compared it with the traditional LR models when producing an LSM of the Three Gorges Reservoir Area. Among the six evaluation indicators considered in their study, the GWR model achieved the best outcome [11]. In the following year, Hong et al. plotted a zoning plan for an LSM in Xingguo County, Shanxi Province using the GWR model and compared it with the traditional LR and SVM models. The results indicated that the GWR model had the highest success rate and prediction accuracy [12]. In the same year, Matsche studied the western part of Oregon and determined that the precision of the GWR model was 6.2% higher than that of the LR model [13]. The use of only the GWR model as an ordinary LSM prediction model improved the LSM to some extent, and enabled spatial scale problems to be considered in an LSM study, but it failed to reveal the essence of the spatial imbalance of the LSM.

In this study, we quantitatively expressed the spatial scale concept of an LSM when studying the spatial scale problem, introduced the concept of spatial scale into the study of LSMs, and built a GWR—particle swarm optimization—SVM (GWR-PSO-SVM) coupled model, to determine the root cause of the impact of spatial scale on an LSM. The aim was to explain the spatial imbalance problem of an LSM, and improve its scientific applicability, accuracy, and reliability.

The remainder of this paper is organized as follows. Section 2 describes the study area and data used in this work. Section 3 reviews the algorithms and model used in this work. Section 4 presents the process used to establish the GWR-PSO-SVM coupled model. Section 5 reports the experimental results, including a comparison between the traditional PSO-SVM coupled models and our new model. Section 6 is a discussion of our model and the final section presents our concluding remarks.

## Study area and data sources

### Study area

In this work, the Zigui-Badong in the Three Gorges Reservoir Area was used as a study area (Fig 1). In terms of topography and geomorphology, the study area is located in the eastern part of the two natural geography units of the Three Gorges Reservoir Area. The area is a basin, and the topography along the river has the characteristics of being low in the middle and high on both banks. In terms of geology, the strata in the study area are fully developed, and only the Lower Devonian, the Upper Silurian and Carboniferous, most of the Cretaceous and a small amount of Tertiary strata are deficient (Fig 2) [14]. Geological disasters occur frequently in the study area, with landslides being the most prominent type of geological disaster. There have been 202 proven landslides in the study area, covering a total area of 23.4 km2, accounting for 6.03% of the entire study area [15].

### Data Source

The following data were used in this study:

➢ 1: 10,000-scale landslide hazard map [15].

➢ Landsat-8 operational land imager (OLI+) sensor data, acquired on 15 September 2013, with a path/row number of 127/39 (https://earthexplorer.usgs.gov/);

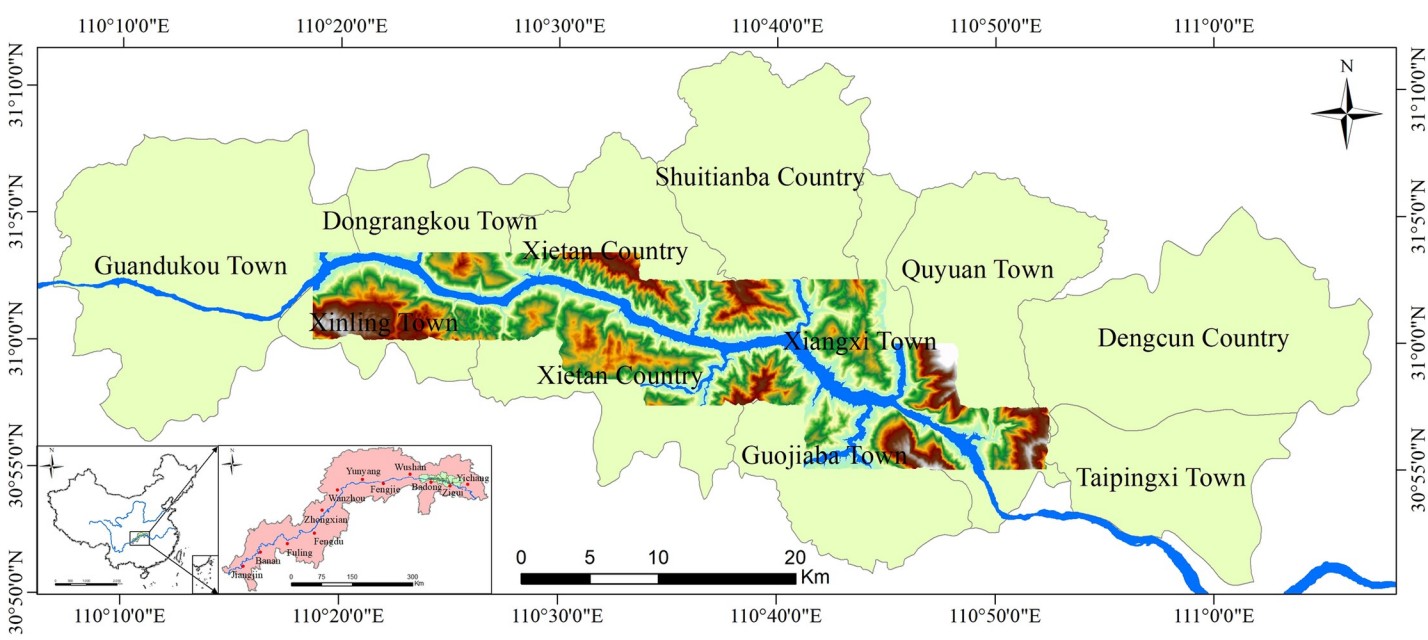

**Fig 1. Geographical location of the study area.**

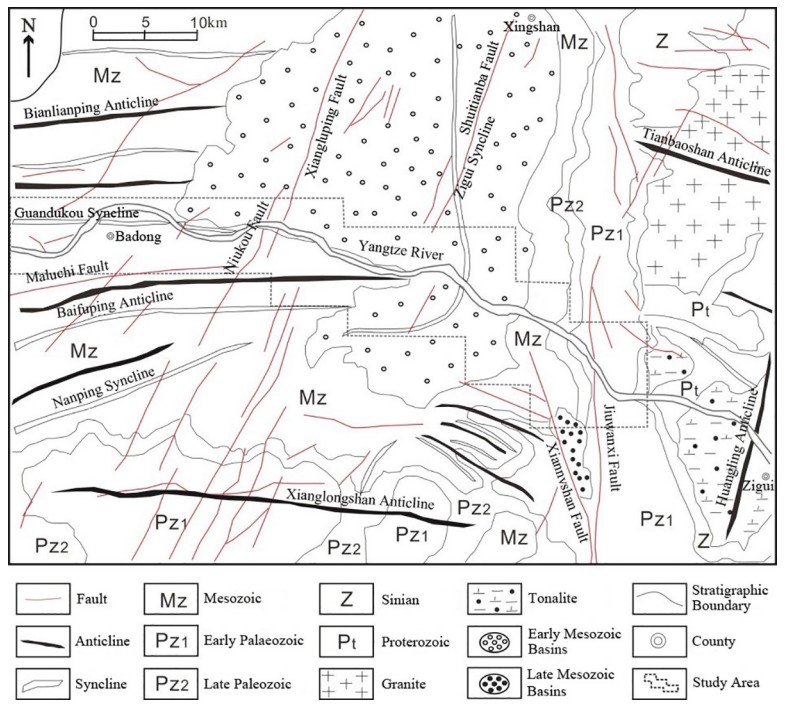

**Fig 2. Geological map of the study area.**

➢ Advanced Spaceborne Thermal Emission and Reflection Radiometer Global Digital Elevation Model (ASTER GDEM) data (https://lpdaac.usgs.gov/tools/data-pool/);

➢ 1: 50,000-scale topographic map and 1: 50,000-scale geological map [14];

➢ Seismic activity data and atmospheric rainfall data from the China Earthquake Administration (CEA, http://www.ceic.ac.cn/history) and the China Meteorological Administration (CMA, http://data.cma.cn/data/cdcdetail/dataCode/SURF_CLI_CHN_MUL_DAY_V3.0.html).

The spatial resolution of the remote sensing (RS) data and the GDEM data was 30 m, and the 1: 10,000-scale landslide hazard map, the 1: 50,000-scale topographic and geological maps could match these data in terms of spatial resolution. The seismic activity and atmospheric rainfall were point data, which had a temporal resolution but no spatial resolution.

## Methods

### The GWR model

Fotheringham et al. first proposed GWR as a method to study the quantitative relationship between two or more variables with spatial distribution characteristics using the regression principle [16]. Local features are used as weights to change the multicollinearity in the global regression model [17, 18]. The related functions are defined as follows:

$$y_i = \beta_0(u_i, v_i) + \sum_{k=1}^{Q} \beta_k(u_i, v_i)x_{ik} + \varepsilon_i, \ i = 1, 2, \cdots, L \tag{1}$$

where $(u_i, v_i)$ are the spatial coordinates of the $i$-th sample; $L$ and $Q$ are the number of samples and regression coefficients, respectively; $y_i$ is the independent variable of the function at point $i$, $x_{ik}$ is the value of the $k$-th explanatory variable of point $i$; $\beta_k(u_i,v_i)$ is the local regression parameter of the $k$-th explanatory variable of point $i$; and $\beta_0(u_i,v_i)$ is the intercept parameter of point $i$. The least squares estimate for $\beta_i$ is as follows:

$$\hat{\beta}_i = (X^T W_i X)^{-1} X^T W_i Y \tag{2}$$

The variance is:

$$var(\hat{\beta}) = (X^T W_i^{-1} X)^{-1} \tag{3}$$

where $W_i$ is a diagonal matrix of $n$ dimension, which is called the spatial weight matrix and is the core of the GWR model. The value on the diagonal is the geographic weight:

$$W_i = \begin{bmatrix} W_{i1} & 0 & \cdots & 0 \\ 0 & W_{i2} & \cdots & 0 \\ \vdots & \vdots & \ddots & \vdots \\ 0 & 0 & \cdots & W_{in} \end{bmatrix} \tag{4}$$

The $W_i$ is chosen based on the choice of kernel function, and the selection of the spatial weight function has a large influence on the parameter estimation of the GWR model.

### The SVM model

The SVM model was first proposed by Vapnik [19]. The model, established on the basis of the Vapnik–Chervonenkis dimension theory and structural risk minimization principle, has

many unique advantages in solving small sample, nonlinear, and high-dimensional pattern recognition problems [20, 21]. Its function is defined as follows:

$$\begin{cases} \min \dfrac{1}{2}\|w\|^2, \\ s.t., \ y_i(w \cdot x_i + b) \geq 1 \end{cases} \tag{5}$$

where $x_i$ is a point on the hyperplane; $y_i$ is the classification marker, $i = 1,2,\cdots,R$; $R$ is the number of samples; $w$ is a vector perpendicular to the hyperplane; $b$ is a constant that is applied to prevent the hyperplane from passing the origin of the coordinate axis; and $\|w\|$ is the 2- norm of $w$. When formula (5) introduces a non-negative slack variable $\xi_i$, a penalty factor $C$ must be introduced to represent the distance from a misclassified point to its correct position. Therefore, the formula (5) can be expressed as:

$$\begin{cases} \min \dfrac{1}{2}\|w\|^2 + C\displaystyle\sum_{i=1}^{n}\xi_i, \\ s.t., \ y_i(w \cdot x_i + b) \geq 1 - \xi_i \end{cases} \tag{6}$$

The RBF can be selected as the kernel function of the SVM, and is used to map the vector of the low-dimensional space into the high-dimensional characteristic space for classification. The function is expressed as:

$$K(x_i, x_j) = \exp(-\gamma\|x_i - x_j\|^2) \tag{7}$$

where, $\gamma$ is the kernel parameter of different radial basis functions.

## The PSO algorithm

The performance of the SVM model relies heavily on two parameters, the penalty factor C and the kernel parameter $\gamma$. The most common method for selecting these two parameters is to use the PSO algorithm to find the optimal solution of the model. Eberhart and Kennedy first proposed the PSO as an intelligent optimization algorithm that mimics bird foraging [22–24]. Its function form is:

$$\begin{cases} V_i^{n+1} = t \cdot V_i^n + c_1 \cdot r_1 \cdot (p_i^n - x_i^n) + c_2 \cdot r_2 \cdot (p_g^n - x_i^n) \\ x_i^{n+1} = x_i^n + V_i^n \end{cases} \tag{8}$$

where $i = 1,2,\cdots,K$; $K$ is the number of particles; $n$ is the current number of iterations; $t$ is the inertia weight; $p_i^n$ is individual optimal position of the $i$-th particle; $p_g^n$ is the optimal position of all particles in the $n$-th iteration; $V_i^n$ and $x_i^n$ are the velocity and position of the $i$-th particle in the $n$-th iteration; $V_i^{n+1}$ and $x_i^{n+1}$ are the speed and position at which the $i$-th particle is updated in the $(n+1)$-th iteration, respectively; $c_1$ and $c_2$ are learning factors; and $r_1$ and $r_2$ are two random numbers between 0 and 1.

## Evaluation models

**Specific category accuracy analysis.** The specific category accuracy analysis method is an improved quantitative analysis method [25]. In this study, the specific category accuracy method considers the number of slope units in the prediction regions. It can be expressed as:

$$p_i = \frac{A_i}{B_i} \cdot 100\% \tag{9}$$

where, $i = 1,2,\cdots,n$; $n$ is the classification number of landslide-prone zonings; $A_i$ is the number of slope units occupied by landslides in $i$-th landslide susceptibility zoning classification; $B_i$ is the number of the slope units in $i$-th landslide susceptibility zoning classification; and $P_i$ is the specific category accuracy in the $i$-th landslide susceptibility zoning classification.

**Overall prediction accuracy analysis.**   The overall prediction accuracy analysis is a commonly used evaluation method for the construction of an LSM. In this study, the original formula was rewritten because there were no landslides in some prediction regions. It was expressed as:

$$p = \frac{\sum_{i=1}^{n_{pr}}(a_i + b_i)}{\sum_{i=1}^{n_{pr}} S_i} \cdot 100\% \tag{10}$$

where $i = 1,2,\cdots,n_{pr}$; $n_{pr}$ is the number of prediction regions; $a_i$ is the number of slope units correctly predicted as landslides in the $i$-th prediction region; $b_i$ is the number of slope units correctly predicted as non-landslide areas in the $i$-th prediction region; and $S_i$ is the number of total slope units in the $i$-th prediction region.

**Receiver operating characteristic (ROC) curve analysis.**   Each point on the ROC curve reflects the susceptibility to the same signal stimulus, with the X- axis representing the negative positive rate specificity and the Y- axis representing the true positive rate sensitivity [26, 27]. There are four possible cases for a binary classification problem, as shown in Table 1.

The AUC refers to the area under the ROC curve. It ranges between 0–1 and its value can be used to intuitively evaluate the quality of the classifier.

## The proposed model

### Coupled model for the LSM based on spatial scale segmentation

By taking the spatial autocorrelation of LSM factors as the breakthrough point, this study regarded the GWR coefficients of the LSM factors as the mathematical basis for the segmentation of the study area. Three basic principles were followed to ensure the rationality of segmentation. First, 58 environmental factors were extracted from the data sources, 18 factors were selected as LSM factors after factor screening, and three of the most important factors were used as segmentation factors to segment the study area into 18 prediction regions. Then, the SVM parameters were optimized by the PSO algorithm, and an LSM for each prediction region was obtained. Finally, all the prediction regions were integrated to establish the LSM model with spatial scale segmentation. A flowchart of the coupled model for the LSM based on a spatial scale analysis was established, as shown in Fig 3.

### Selection of LSM calculation units

According to Guzzetti et al., all LSM calculation units can be summarized as either grid cells, geographic units, unique conditional units, slope units, or sub-basin units [28]. In this study,

**Table 1.  The four cases for a binary classification problem.**

| | | Prediction | | Total |
|---|---|---|---|---|
| | | Positive, P | Negative, N | |
| Actual | Positive, P | True Positive, TP | False Negative, FN | Actual Positive, TP+FN |
| | Negative, N | False Positive, FP | True Negative, TN | Actual Negative, FP+TN |
| Total | | Predicted Positive, TP+FP | Predicted Negative, FN+TN | TP+FP+FN+TN |

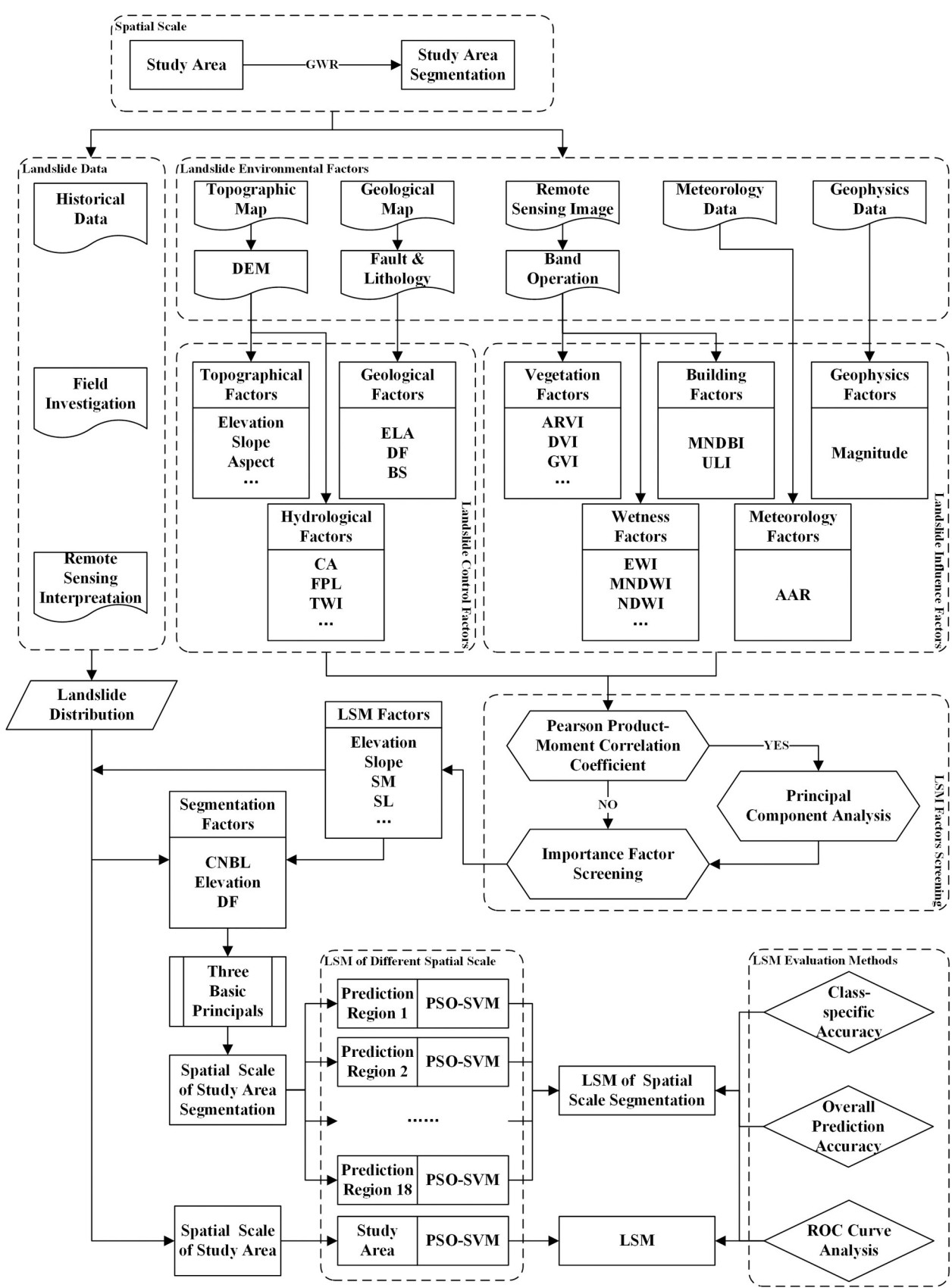

**Fig 3. A flowchart of the establishment of the coupled model for the LSM based on spatial scale segmentation.** Abbreviations in this figure: GWR = geographically weighted regression, PSO = particle swarm optimization, SVM = support vector machine, LSM = landslide susceptibility map, ROC = receiver operation characteristic, DEM = digital elevation model, CA = catchment area, FPL = flow path length, TWI = topographic wetness index, ELA = engineering lithologic assemblage, DF = distance to fault, BS = bedding structure, ARVI = atmospherically resistant vegetation index, DVI = difference vegetation index, GVI = green vegetation index, EWI = enhanced water index, MNDWI = modified normalized difference water index, NDMI = normalized difference moisture index, MNDBI = modified normalized difference building index, ULI = urban land-use index, AAR = average annual rainfall, SM = slope morphology, SL = slope length, CNBL = channel network basic level, DR = distance to river.

the slope unit was selected as the LSM calculation unit. After calculation and modification, a total of 2,790 slope units were obtained in the study area, of which the smallest area was 11,823.9 $m^2$ and the largest was 819,444 $m^2$.

## Screening of LSM factors

In this work, through an analysis of historical landslide data and a summary of previous research in the study area, the LSM factors were divided into two categories: control factors and influencing factors. The control factors included geomorphological, geological, and hydrological factors. The influencing factors included surface cover index, geophysical, and meteorological factors. In this study, based on the geological and topographic maps, RS image data, field survey reports, and other data, a total of 58 LSM factors in two categories and eight sub-categories were extracted by RS and geographic information system. This is summarized in Table 2.

Some of these 58 LSM factors were obtained by the DEM, and had a large correlation with each other. Therefore, not all of these 58 factors were involved in the modeling and calculation of LSM, but they needed to be further analyzed and screened. There were two main steps in the analysis.

**Pearson product-moment correlation coefficient (PPMCC) analysis and principal component analysis (PCA).** In this study, a PPMCC analysis was used to analyze the correlations among the LSM factors of five sub-categories (geomorphology, hydrology, vegetation index, wetness index, and building index), and the factors with a significant correlation were deleted [29].

**Table 2. The initial landslide susceptibility evaluation factors.**

| Categories | Sub-categories | Factors |
|---|---|---|
| Control Factors | Geomorphology | Elevation, Slope, Aspect, Slope Morphology (SM), Terrain Ruggedness Index (TRI), Slope Length (SL), Plan Curvature (PLC), Profile Curvature (PRC), Terrain Surface Texture (TST), Terrain Surface Convexity (TSC), Topographic Position Index (TPI), TPI based Landform Classification (TPILC), Topographic Convergence Index (TCI), Cross-Sectional Curvature (CSC), General Curvature (GC), Longitudinal Curvature (LC), Tangential Curvature (TAC), Maximum Curvature (MAXC), Minimum Curvature (MINC), Mid-slope Position (MSP), Total Curvature (TOC), Slope Height (SH), Valley Depth (VD) |
| | Geology | Engineering Lithologic Assemblage (ELA), Distance to Fault (DF), Bedding Structure (BS) |
| | Hydrology | Catchment Area (CA), Flow Path Length (FPL), LS Factor (LSF), Melton Ruggedness Number (MRN), Topographic Wetness Index (TWI), Distance to River (DR), Catchment Slope (CS), Channel Network Basic Level (CNBL), Floe Width (FW), Stream Power Index (SPI), Terrain Classification Index for Lowlands (TCIL), Vertical Distance to Channel Network (VDCN), Flow Line Curvature (FLC) |
| Influence Factors | Vegetation Index | Atmospherically Resistant Vegetation Index (ARVI), Difference Vegetation Index (DVI), Green Vegetation Index (GVI), Normalized Difference Water Index (NDWI), Normalized Difference Vegetation Index (NDVI), Ration Vegetable Index (RVI), Transformed Vegetable Index (TVI), Fractional Vegetation Cover (FVC) |
| | Wetness Index | Enhanced Water Index (EWI), Modified Normalized Difference Water Index (MNDWI), Normalized Difference Moisture Index (NDMI), Normalized Difference Water Index (NDWI), Ratio Moisture Index I (RMI1), Ratio Moisture Index II (RMI2), The Wetness Index of the Tasseled Cap (WITC) |
| | Building Index | Modified Normalized Difference Building Index (MNDBI), Urban Land-use Index (ULI) |
| | Geophysics | Magnitude |
| | Meteorology | Average Annual Rainfall (AAR) |

**Table 3. Final landslide susceptibility evaluation factors after screening.**

| Categories | Sub-Categories | Factors | Units | Ranges |
|---|---|---|---|---|
| Control Factors | Geomorphology | Elevation | m | 80.00~2,000.00 |
| | | Slope | ° | 0.00~78.42 |
| | | SM | - | (1) V/V; (2) GE/V; (3) X/V; (4) V/GR; (5) GE/GR; (6) X/GR; (7) V/X; (8) GE/X; (9) X/X |
| | | SL | m | 0~3,938.24 |
| | | TST | - | 0.00~0.69 |
| | | PCCFC-1 | - | 173.06~573.32 |
| | Geology | ELA | - | (1) mudstone, shale and Quaternary deposits; (2) sandstones and thinly bedded limestones; (3) limestones and massive sandstones |
| | | DF | m | 0~8 739.89 |
| | | BS | - | (1) over-dip slope; (2) under-dip slope; (3) dip-oblique slope; (4) transverse slope; (5) anaclinal-oblique slope; (6) anaclinal slope [31] |
| | Hydrology | MRN | - | 0~42.292 |
| | | TWI | - | 4.442~18.03 |
| | | DR | m | 377.32~4,562.34 |
| | | CNBL | m | 80.23~1,353.91 |
| Influence Factors | Vegetation Index | PCVIFC-1 | - | 0.00~1.00 |
| | Wetness Index | PCWIFC-1 | - | 0.00~1.00 |
| | Building Index | PCBIFC-1 | - | 0.00~1.00 |
| | Geophysics | Magnitude | $Ms$ | 1.0~5.0 |
| | Meteorology | AAR | mm | 964.03~1,090.24 |

In the three sub-categories other than geomorphology and hydrology, there were strong correlations among multiple factors. In the geomorphology sub-category, the profile curvature, topographic position index (TPI), TPI based landform classification, cross-sectional curvature, general curvature, longitudinal curvature, maximum curvature and minimum curvature factors had strong correlations, and formed a curvature factor combination. The same phenomenon occurred in the vegetation index, wetness index and the building index sub-categories, and formed the vegetation index factor combination, the wetness index factor combination, and the building index factor combination.

To retain the multi-factor effective information and remove the linear correlation of these factor combinations, the PCA method was used [30]. In this study, the first, second, and third principal component of the curvature factor combination (PCCFC-1, 2, 3), the first principal component of the vegetation index factor combination (PCVIFC-1), the first principal component of the wetness index factor combination (PCWIFC-1), and the first principal component of the building index factor combination (PCBIFC-1) were retained. After the PPMCC analysis and PCA, there were 32 factors remaining.

**Factor importance screening based on the SVM model.** In this study, the SVM was used as the prediction model for the LSM. The model can determine the importance of each factor according to the degree of contribution of the LSM. Based on this, this study removed the unimportant factors to improve the efficiency and accuracy of the LSM. After repeated experiments and comparisons, in combination with previous research results and based on the LSM factors that played a major role in most landslide studies, the importance threshold of the LSM factors was determined (0.005). Finally, 18 LSM factors were obtained. This is summarized in Table 3.

### The GWR-based segmentation of the study area

Based on the calculation of the GWR coefficients of each LSM factor, the natural breakpoint method was used for classification in this study [32]. To segment the study area, theoretically, the classification results of all the LSM factors should be superimposed to reduce the spatial autocorrelation for each LSM factor. However, due to the excessive number of LSM factors, the superposition of all LSM factor classification results may generate too many small areas, and have a great impact on the subsequent steps. Moreover, too many segmentation areas may also make the spatial distance between the areas smaller, in turn increasing the spatial autocorrelation of the LSM factors. After repeated studies, three basic principles were identified that should be followed when the GWR model was used for spatial scale segmentation:

➢ Select the same appropriate number of classifications for all LSM factors;

➢ Select only the most important LSM factors as the segmentation factors to segment the study area;

➢ In light of the results of spatial scale segmentation, regions that are too small should be merged into adjacent regions, and the integrity of the landslide surface should be guaranteed.

Based on principles 1 and 2, we selected the three most important LSM factors in the SVM model (channel network basic level (CNBL), elevation, and distance to river) for use as the segmentation factors. Each segmentation factors was then divided into two categories by the natural breakpoint method. The final result of the spatial scale segmentation of the study area was superimposed and processed by principle 3, with a total of 18 small areas, which were called 18 prediction regions. The segmentation process is shown in Fig 4.

### Establishment of an LSM model based on GWR

For each prediction region, all the landslide slope units and randomly selected non-landslide slope units constituted a training data set (at a 1: 1 ratio) to conduct training of the PSO-SVM coupled model. All the slope units in the region, as the verification sample data set, were input into the trained coupled model, and an LSM was obtained for each prediction region. The optimal solution of the PSO-SVM coupled model for each prediction region is shown in Table 4.

## Results

### Experimental results of the GWR-PSO-SVM coupled model

The LSMs of all prediction regions were combined to obtain an LSM based on spatial scale segmentation, i.e., the LSM of the GWR-PSO-SVM coupled model. The landslide susceptibility index (LSI) is a form of LSM, which is a continuous value from 0 to 1. This is shown in Fig 5.

### Establishment of a comparative experiment to test the PSO-SVM coupled model

To compare the precision and accuracy of the GWR-PSO-SVM coupled model proposed in this study, and especially to verify the correctness of the study area spatial scale segmentation using the GWR method, a comparative experiment was conducted. To verify the influence of spatial imbalance on the LSM, the PSO-SVM coupled model was used in the comparative experiment. The operational process of the PSO-SVM coupled model was basically the same as that of the GWR-PSO-SVM coupled model, with just the spatial scale segmentation using

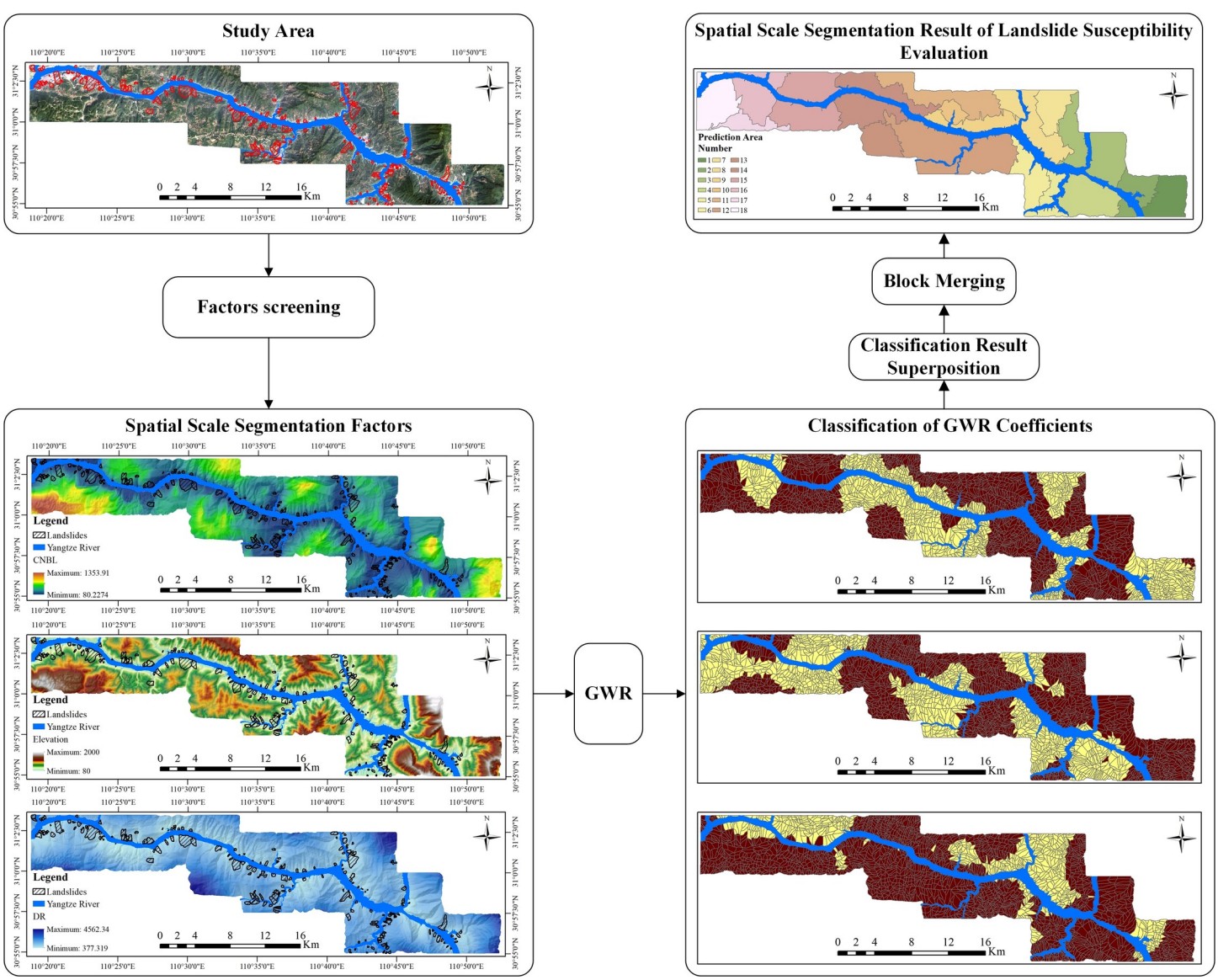

**Fig 4. The process of spatial scale segmentation based on GWR.**

**Table 4. Optimal solutions of the PSO-SVM coupled model for each prediction region.**

| Prediction Regions ID | $C$ | $\gamma$ | Prediction Regions ID | $C$ | $\gamma$ |
|---|---|---|---|---|---|
| 2 | 16 | 1 | 10 | 4 | 0.5 |
| 3 | 8 | 2 | 12 | 8 | 0.25 |
| 4 | 8 | 0.125 | 14 | 4 | 4 |
| 5 | 2 | 4 | 15 | 8 | 0.25 |
| 6 | 8 | 2 | 16 | 4 | 0.125 |
| 8 | 8 | 1 | 18 | 8 | 1 |
| 9 | 4 | 4 | - | - | - |

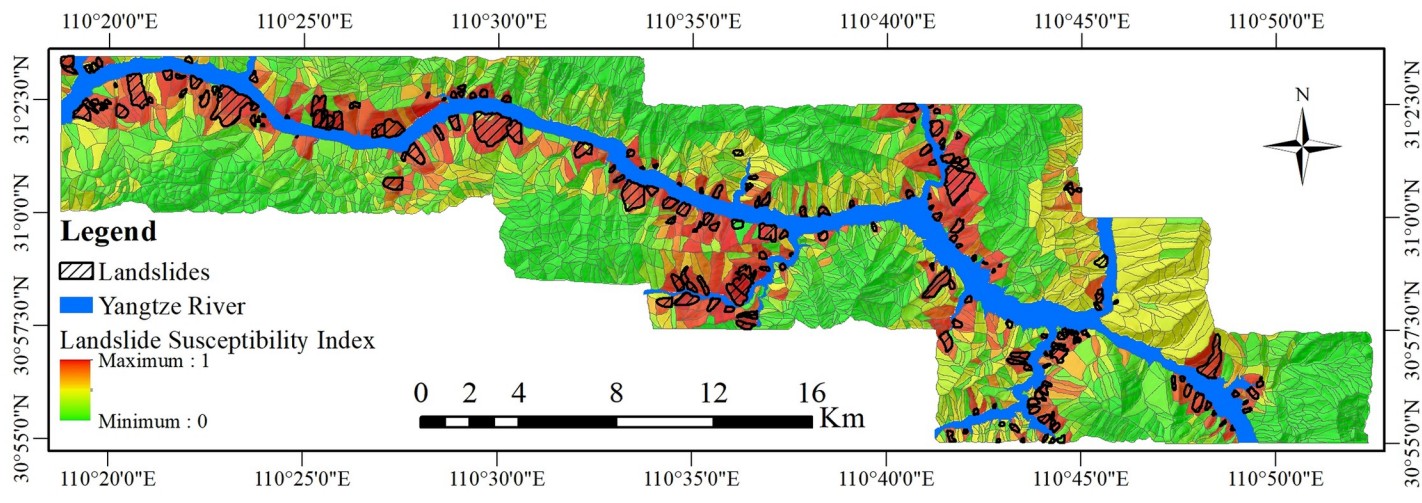

**Fig 5. The landslide susceptibility index (LSI) produced by the GWR-PSO-SVM coupled model.**

GWR coefficients removed, and the selection of the LSM factors were consistent with those selected for the GWR-PSO-SVM coupled model. The PSO algorithm determined that the optimal solutions for C and γ in the SVM model were 4 and 1, respectively, and the LSM for the PSO-SVM coupled model was obtained. This is shown in Fig 6.

To increase the readability of the LSM, the fixed threshold method was used in this study. Values of 0.1, 0.3, 0.7, and 0.9 were selected as the classification thresholds. The LSI was divided into five categories to obtain the landslide susceptibility zoning (LSZ): very low susceptibility areas, low susceptibility areas, medium susceptibility areas, high susceptibility areas, and very high susceptibility areas. The LSZs from the two experiments are shown in Fig 7 and Fig 8.

## Evaluation model results and analysis

**Specific category accuracy analysis.** The specific category accuracy results of the two experiments were calculated using formula (9) and are shown in Table 5.

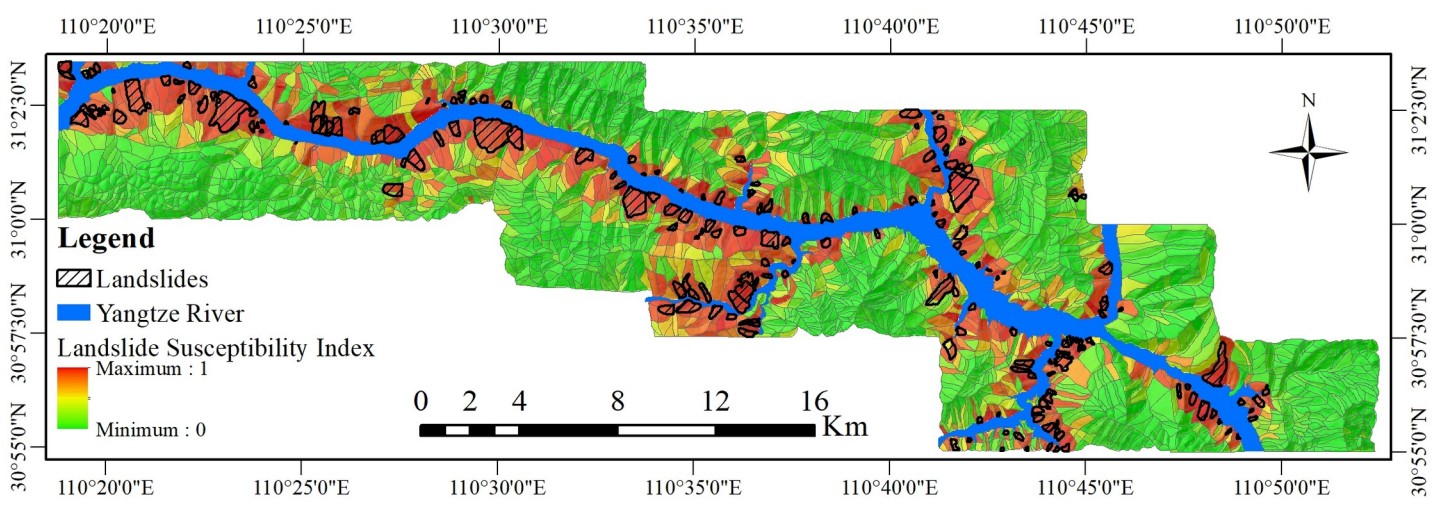

**Fig 6. The LSI produced by the PSO-SVM coupled model.**

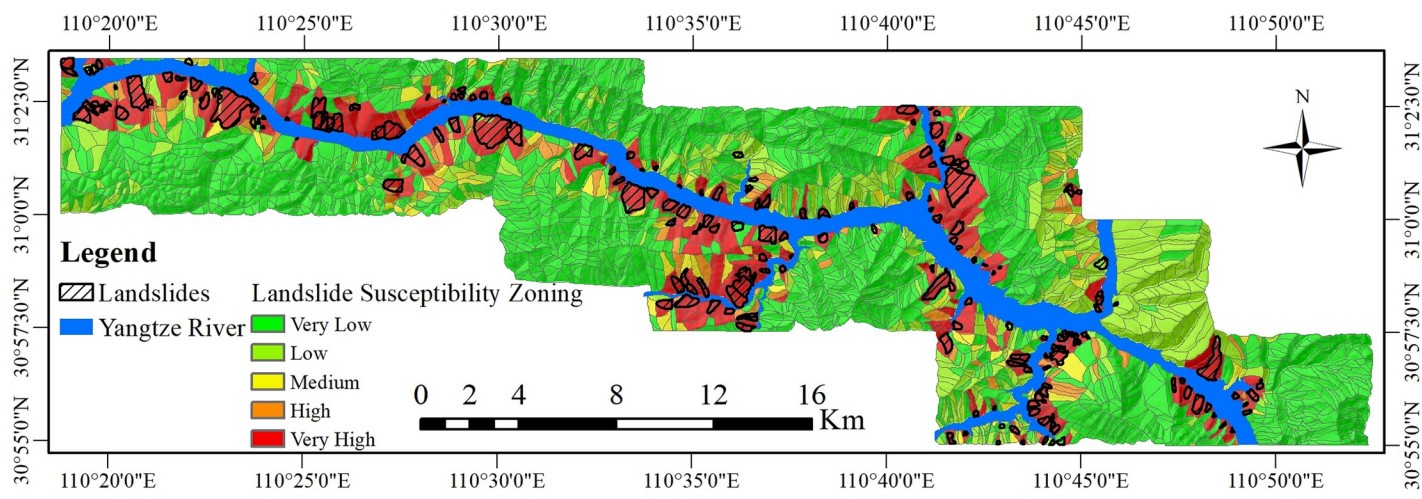

**Fig 7. The landslide susceptibility zoning (LSZ) based on the GWR-PSO-SVM coupled model.**

The results in Table 5 show that the GWR-PSO-SVM coupled model identified more slope units in the "Very High" LSZ category (85.75%) than the PSO-SVM coupled model (68.35%). The GWR-PSO-SVM coupled model was significantly superior to the PSO-SVM coupled model.

**Overall prediction accuracy analysis.** The overall prediction accuracy analysis results of the two experiments are shown in Table 6.

It can be clearly seen from Table 6 that the overall prediction accuracy of the PSO-SVM coupled model was 84.44%. In the GWR-PSO-SVM coupled model, the prediction accuracy of most prediction regions was greater than that of the PSO-SVM coupled model. The overall prediction accuracy of the GWR-PSO-SVM coupled model was 87.86%, which was more accurate than the PSO-SVM coupled model.

**The ROC curve analysis.** In this study, the ROC curve was constructed using the real data of each slope unit as the state variable, and the LSMs at different spatial scales as the test variable, as shown in Fig 9.

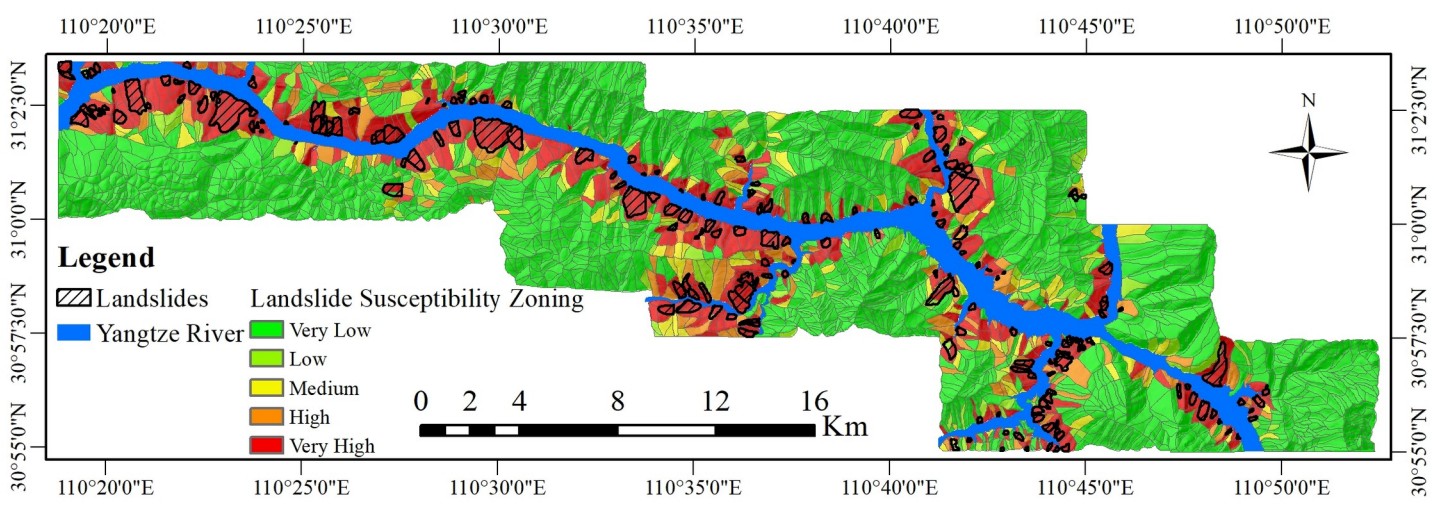

**Fig 8. The landslide susceptibility zoning (LSZ) based on the PSO-SVM coupled model.**

**Table 5. Specific category accuracy analysis results for the two experiments.**

| LSZ | GWR-PSO-SVM | | | PSO-SVM | | |
|---|---|---|---|---|---|---|
| | Number of Slope Units for Landslides | Number of Slope Units | Specific Category Accuracy | Number of Slope Units for Landslides | Number of Slope Units | Specific Category Accuracy |
| Very Low | 8 | 1,439 | 0.56% | 7 | 1,615 | 0.43% |
| Low | 3 | 281 | 1.07% | 9 | 260 | 3.46% |
| Medium | 37 | 567 | 6.53% | 20 | 251 | 7.97% |
| High | 21 | 103 | 20.39% | 37 | 168 | 22.02% |
| Very High | 343 | 400 | **85.75%** | 339 | 496 | **68.35%** |

It can be seen from Fig 9 that both of the experiments produced good results. However, in the ROC curve of the GWR-PSO-SVM coupled model, the closest point to the upper left corner was farther from the reference line than for the PSO-SVM coupled model. A qualitative analysis showed that the result of the GWR-PSO-SVM coupled model was better.

For a quantitative analysis, the AUC calculations for the two experiments are shown in Table 7.

As shown in Table 7, the AUC value of the GWR-PSO-SVM coupled model was 0.965, i.e., greater than the value of 0.944 for the PSO-SVM coupled model, indicating that in the ROC curve analysis, the result for the GWR-PSO-SVM coupled model was better than that for the PSO-SVM coupled model.

## Discussion

Based on previous analyses and LSM characteristics, there were four main reasons for the differences among LSMs: (1) the spatial scale of LSMs; (2) the factors used in the construction of LSMs;(3) the calculation unit used to the construct the LSMs; and (4) the prediction model used to construct the LSMs. In most LSM studies, the factors, calculation units, and prediction

**Table 6. Overall prediction accuracy analysis results for the two experiments.**

| Experiments | Prediction Region [1] | Number of Correct Predictions | Number of Total Predictions | Prediction Accuracy [2] |
|---|---|---|---|---|
| GWR-PSO-SVM | 2 | 82 | 93 | 88.17% |
| | 3 | 159 | 185 | 85.95% |
| | 4 | 125 | 138 | 90.58% |
| | 5 | 76 | 91 | 83.52% |
| | 6 | 104 | 126 | 82.54% |
| | 8 | 131 | 144 | 90.97% |
| | 9 | 94 | 104 | 90.38% |
| | 10 | 34 | 38 | 89.47% |
| | 12 | 352 | 399 | 88.22% |
| | 14 | 272 | 310 | 87.74% |
| | 15 | 241 | 279 | 86.38% |
| | 16 | 101 | 109 | 92.66% |
| | 18 | 111 | 126 | 88.10% |
| | Total | 1,882 | 2,142 | **87.86%** |
| PSO-SVM | Study Area | 2,356 | 2,790 | **84.44%** |

[1] In this table, the prediction regions of the GWR-PSO-SVM coupled model did not contain the prediction regions without landslides.

[2] Represents the prediction accuracy in each prediction region, with the overall prediction accuracy of the two experiments marked in bold.

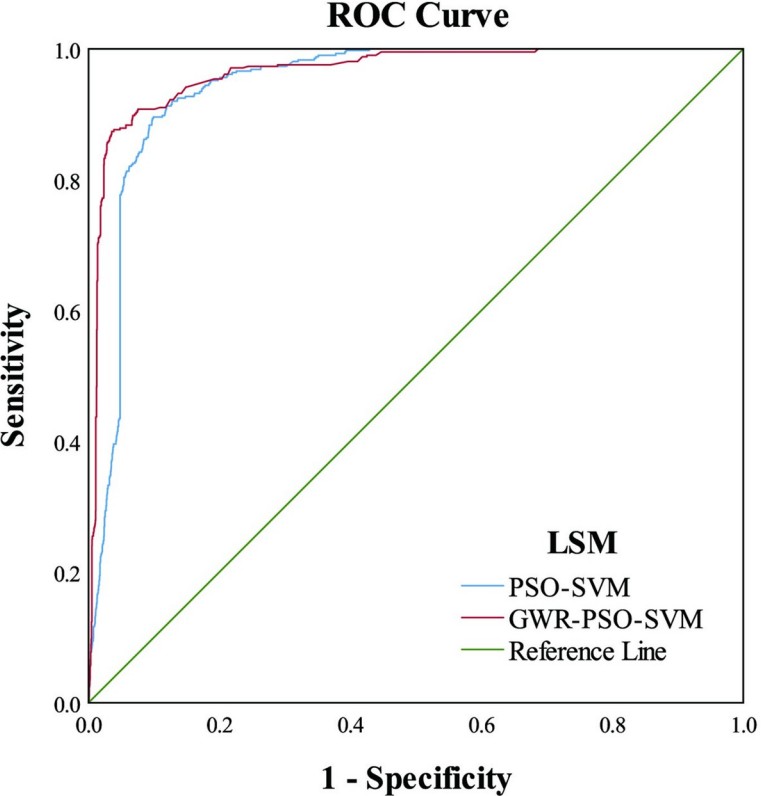

**Fig 9. The ROC curves for the two experiments.**

models have been considered to be the main reasons for the differences among LSMs. Although these are not the same points considered in this work, this is not a contradictory position.

At the same spatial scale, the main reasons leading to the differences among LSMs were derived from the factors, calculation units, and prediction models. However, as research has intensified and with the introduction of spatial scale problems, the spatial scale, factors, calculation units, and prediction models have been identified as the root causes of the differences among LSMs.

In the actual experiments, the LSMs obtained from large areas were often different from and even the opposite of those obtained from a smaller area inside the large area when the same factors, calculation unit, and prediction model were used.

However, many of the LSM prediction models used previously were not originally based on geology or geography, but evolved from economics, statistics, and other disciplines. Therefore,

**Table 7. Area under the curve (AUC) analysis for the two experiments.**

| Test Result Variable(s) | Area | Std. Error [a] | Asymptotic Sig. [b] | Asymptotic 95% confidence Interval | |
|---|---|---|---|---|---|
| | | | | Low Bound | Upper Bound |
| PSO-SVM | 0.944 | 0.005 | 0.000 | 0.934 | 0.953 |
| GWR-PSO-SVM | 0.965 | 0.005 | 0.000 | 0.956 | 0.974 |

[a] Under the nonparametric assumption.

[b] Null hypothesis: true area = 0.5.

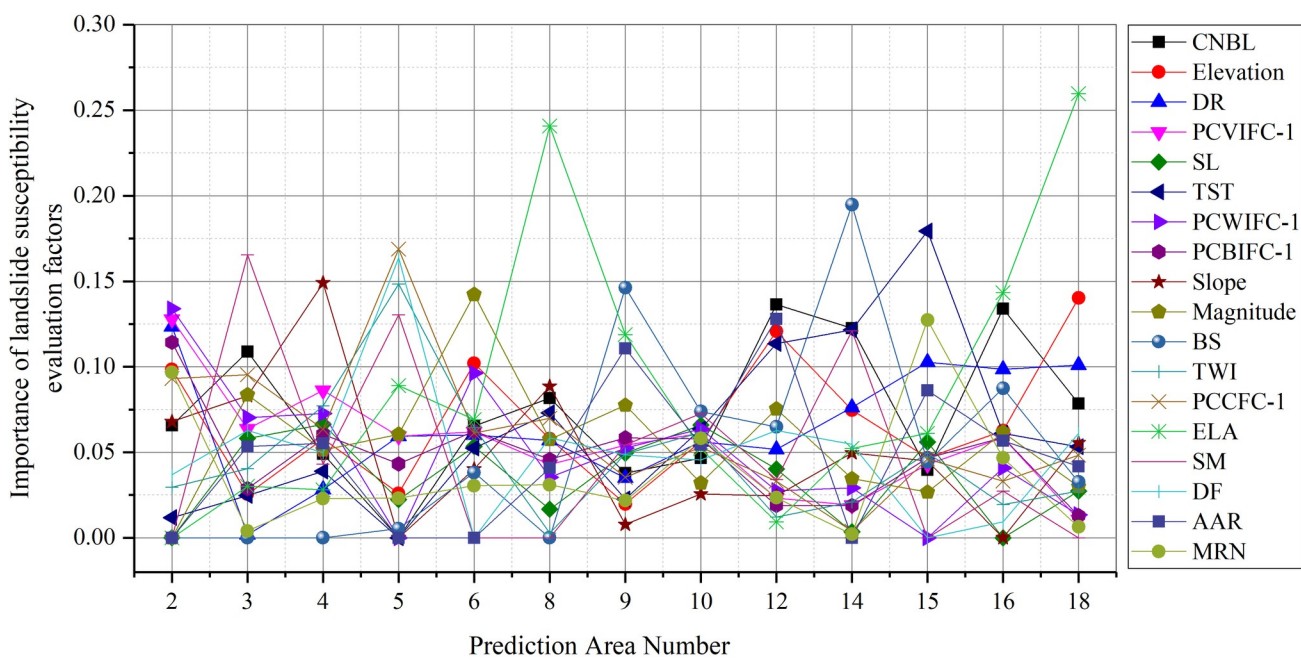

**Fig 10. Schematic diagram of the changes in the significance of LSM factors in each prediction region based on the GWR-PSO-SVM coupled model.**

these prediction models have been subjected to repeated verifications over several years or even decades, and have been shown to have objectivity, applicability, and stability. Although there are five kinds of calculation unit, they are all fundamentally based on the grid unit. The grid unit is determined by the mathematical and physical properties of remote sensing satellite images, which also have objectivity. Considering this situation, this study focused on the LSM factors and spatial scale.

A total of 13 experiments using the GWR-PSO-SVM coupled model were completed in this study, and in each experiment, each LSM factor had a different importance, as shown in Fig 10. For the convenience of comparison, the order of factors in the legend was arranged from high (0.241) to low (0.005) according to the importance score of LSM factors in the SVM model.

The following results can be observed from Fig 10:

The important factors in the PSO-SVM coupled model did not have importance (i.e., the value was 0) in some prediction regions of the GWR-PSO-SVM coupled model. In prediction region 5, for instance, the most important LSM factor in the PSO-SVM coupled model (CNBL) had no importance. There were significant differences in the importance of LSM factors at different spatial scales.

After the study area was segmented, the figure shows that even the adjacent regions 2 and 3 had different importance rankings for the LSM factors, indicating the variable importance of the LSM factors in different prediction regions, and illustrating that the LSM factors had regional characteristics.

## Conclusion

Using Zigui-Badong in the Three Gorges Reservoir Area as a case study, the GWR model was coupled with the PSO-SVM model to utilize the advantages of GWR in the processing of

spatial heterogeneity. According to the GWR coefficients of LSM factors, the study area was divided into several prediction regions to solve the problem of spatial imbalances in an LSM. To verify the effectiveness of the proposed method, the outcomes of the GWR-PSO-SVM coupled model and the PSO-SVM coupled model were compared using three evaluation methods: specific category accuracy analysis, overall prediction accuracy analysis, and AUC analysis. The results for the GWR-PSO-SVM coupled model for these three evaluation methods were 85.75%, 87.86%, and 0.965, respectively, while the results for the traditional PSO-SVM coupled model were 68.35%, 84.44%, and 0.944, respectively. Comparing the three evaluation methods, the results for the GWR-PSO-SVM coupled model were 17.4%, 3.42%, and 0.021 higher than those of the PSO-SVM coupled model, respectively, and the new model had obvious advantages over the former model.

It was found that the importance of LSM factors in different areas were actually different. The method in which LSM factors were statistically calculated and assigned a weighted value by the prediction model for a complete study area was obviously questionable.

The spatial scale of the study area essentially affects the importance of LSM factors. Therefore, based on the LSM factors and GWR model, the spatial scale segmentation method of the study area developed in this study that was obtained by the selection of regional segmentation factors, calculation and classification of GWR coefficients, superposition of the classification results, and human-computer interaction modification was an effective method to solve this problem.

## Acknowledgments

We are grateful to the Headquarters of Prevention and Control of Geo-Hazards in the Area of the Three Gorges Reservoir for providing data and material. We also thank the editor and anonymous referees for their comments.

## Author Contributions

**Conceptualization:** Xianyu Yu.

**Data curation:** Xianyu Yu.

**Funding acquisition:** Xianyu Yu.

**Investigation:** Huachen Gao.

**Methodology:** Xianyu Yu, Huachen Gao.

**Software:** Huachen Gao.

**Supervision:** Xianyu Yu.

**Writing – original draft:** Xianyu Yu, Huachen Gao.

**Writing – review & editing:** Xianyu Yu.

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
