## [Decision Letter · Decision Letter 0]

26 Nov 2019

PONE-D-19-27136

A landslide susceptibility map based on spatial scale segmentation: a case study at zigui-badong in the three gorges reservoir area, China

PLOS ONE

Dear Dr. Yu,

Thank you for submitting your manuscript to PLOS ONE. After careful consideration, we feel that it has merit but does not fully meet PLOS ONE’s publication criteria as it currently stands. Therefore, we invite you to submit a revised version of the manuscript that addresses the points raised during the review process.

We would appreciate receiving your revised manuscript by Jan 10 2020 11:59PM. To enhance the reproducibility of your results, we recommend that if applicable you deposit your laboratory protocols in protocols.io, where a protocol can be assigned its own identifier (DOI) such that it can be cited independently in the future. For instructions see: http://journals.plos.org/plosone/s/submission-guidelines#loc-laboratory-protocols

We look forward to receiving your revised manuscript.

Kind regards,

Claudionor Ribeiro da Silva

Academic Editor

PLOS ONE

Journal Requirements:

1. Please clarify in your data source section how the data can be accessed by future researchers, e.g. by providing a citation to appropriate literature, a URL link to a publically available database, or a sentence clarifying which organisation data can be requested from.

Reviewers' comments:

Reviewer's Responses to Questions

**Comments to the Author**

1. Is the manuscript technically sound, and do the data support the conclusions?

Reviewer #1: Partly

Reviewer #2: Yes

Reviewer #3: Yes

2. Has the statistical analysis been performed appropriately and rigorously? 

Reviewer #1: Yes

Reviewer #2: Yes

Reviewer #3: Yes

3. Have the authors made all data underlying the findings in their manuscript fully available?

Reviewer #1: No

Reviewer #2: Yes

Reviewer #3: Yes

4. Is the manuscript presented in an intelligible fashion and written in standard English?

Reviewer #1: No

Reviewer #2: Yes

Reviewer #3: No

5. Review Comments to the Author

Reviewer #1: The topic of the manuscript is interesting, but the manuscript is poorly organized. It looks like a textbook. There are not clear research questions and hypothesis for testing. Detail discussion lacks.

Reviewer #2: The manuscript is well-written, balanced and presents an important contribution to the current scientific knowledge. Hence, as such it should be published, maybe even as it is.

Methods follow standard scientific ways of research. Computational aspect of this study is amazing. The SVM model and PSO algorithm are very informative but maybe they could be presented in a bit more detail, if authors share my opinion, but it is not mandatorily. I understand that the references are given but maybe some of readers would like to have these information given in paper, merely to ease the reading; but again am not imposing it. What I really have to commend on this occasion is the ROC analysis. This so marvelous tool that I intend to use in my future research, so I will take this opportunity to personally thank authors on that. Results are presented in a standard manner and clear. The following discussion again maybe could be more detailed but not necessarily so. Personally, I belong to the group of authors that prefer a little bit longer discussions but it is not of some importance here.

According to all of this, I think that with or without mention changes to be made (I will leave it to the authors to decide) this manuscript definitely and irrevocably should be published.

Reviewer #3: In terms of format, this paper is relatively standard. The research methods, contents and results in this paper are all good.

However, there are some little problems in a small amount of languages .

For example,”large landslides were divided into two categories of small & medium and large & detailed scales, ”. It’s not clear what are the two categories?

6. PLOS authors have the option to publish the peer review history of their article (what does this mean?). If published, this will include your full peer review and any attached files.

Reviewer #1: No

Reviewer #2: Yes: Ivan Grbatinic

Reviewer #3: Yes: Ying Chen is working on Department of Computer Science in Taizhou University, Taizhou, China as an assistant professo.

---

## [Author Response · Author response to Decision Letter 0]

29 Jan 2020

Academic Editor:

We first sincerely thank for your "minor revision" opinion, which is a kind of affirmation for our research. In addition, we would like to thank you for your comment, it is very useful for us to improve the paper. Based on the comment, we have tried our best to improve the manuscript as much as possible. In the following, we will address each comment in detail.

1. Please clarify in your data source section how the data can be accessed by future researchers, e.g. by providing a citation to appropriate literature, a URL link to a publically available database, or a sentence clarifying which organisation data can be requested from.

R: Thanks for your comment. The availability of data in the paper is a serious issue. Our modifications are as follows:

 1: 10,000-scale landslide hazard map [15];

 Landsat-8 operational land imager (OLI+) sensor data, acquired on 15 September 2013, with a path/row number of 127/39 (Download link: https://earthexplorer.usgs.gov/);

 Advanced Spaceborne Thermal Emission and Reflection Radiometer Global Digital Elevation Model (ASTER GDEM) data (Download link: https://lpdaac.usgs.gov/tools/data-pool/);

 1: 50,000-scale topographic map and 1: 50,000-scale geological map [14];

 Seismic activity data and atmospheric rainfall data from the China Earthquake Administration (CEA, download link: http://www.ceic.ac.cn/history) and the China Meteorological Administration (CMA, download link: http://data.cma.cn/data/cdcdetail/dataCode/SURF_CLI_CHN_MUL_DAY_V3.0.html).

The spatial resolution of the remote sensing (RS) data and the GDEM data was 30 m, and the 1: 10,000-scale landslide hazard map, the 1: 50,000-scale topographic and geological maps could match these data in terms of spatial resolution. The seismic activity and atmospheric rainfall were point data, which had a temporal resolution but no spatial resolution.

The landsat-8 data, ASTER GDEM data, seismic activity data and atmospheric rainfall data can be downloaded through the download link. But the 1: 10,000-scale landslide hazard map, 1: 50,000-scale topographic map and 1: 50,000-scale geological map are the confidential in China, we have no way to obtain them online in publicly. All the map data used in this paper are declassified, and we're not allowed to share those data.

I hope you are satisfied with this reply.

Reviewer #1: 

We first sincerely thank the reviewer for the comments, most of which are very useful for us to improve the paper. Meanwhile, we are sorry that some points of the previous version confused the reviewer. Based on the comments, we have tried our best to improve the manuscript as much as possible. In the following, we will address each comment in detail.

1. The topic of the manuscript is interesting, but the manuscript is poorly organized. It looks like a textbook. There are not clear research questions and hypothesis for testing. Detail discussion lacks.

R: First of all, thanks for your affirmation of the topic of my manuscript. 

Meanwhile, I know exactly which section "poorly organization" and "looks like a textbook" refer to in your comments. In fact, I strictly followed the structure requirements of the PLoS One journal for the manuscript, and consciously reduced the description of the existing methods and increased the description of the experiment process. In my opinion, for the landslide susceptibility map, any step of the experimental process, whether the factor selection or the GWR-based segmentation of the study area proposed in this paper, has a huge impact on the results. Therefore, although I have made great efforts to control the description of the experimental process, I still cannot avoid the impression of "textbook" for you at last. I'm very sorry. But I still insist to the structure of my paper. Because I think the experiment in the paper must be repeatable. Other researchers can use their data and my method to get similar conclusions, which is very important.

In my “Introduction” section, I analyzed the deficiencies in the current research from three levels: “the traditional methods ignore the spatiality of landslide susceptibility map”, “some methods weaken the concept of spatiality of landslide susceptibility map” and “the common GWR method fails to find the essence of the imbalance of landslide susceptibility map”. Therefore, my research question is to solve the imbalance of results in landslide susceptibility, which is also mentioned in the "Abstract" section.

Combining my research question, I discussed the obtained experiments and summarized two results. I believe that these two results cannot completely solve the problem of imbalance of the landslide susceptibility map. Limited by my ability, there are many laws, results, and essences that I have not discovered. It is my motivation to keep learning. In addition, it is also the charm of scientific research.

I hope you are satisfied with this reply.

Reviewer #2:

We first sincerely thank the reviewer for the comments, most of which are very useful for us to improve the paper. Meanwhile, we are sorry that some points of the previous version confused the reviewer. Based on the comments, we have tried our best to improve the manuscript as much as possible. In the following, we will address each comment in detail.

1. The manuscript is well-written, balanced and presents an important contribution to the current scientific knowledge. Hence, as such it should be published, maybe even as it is.

Methods follow standard scientific ways of research. Computational aspect of this study is amazing. The SVM model and PSO algorithm are very informative but maybe they could be presented in a bit more detail, if authors share my opinion, but it is not mandatorily. I understand that the references are given but maybe some of readers would like to have these information given in paper, merely to ease the reading; but again am not imposing it. What I really have to commend on this occasion is the ROC analysis. This so marvelous tool that I intend to use in my future research, so I will take this opportunity to personally thank authors on that. Results are presented in a standard manner and clear. The following discussion again maybe could be more detailed but not necessarily so. Personally, I belong to the group of authors that prefer a little bit longer discussions but it is not of some importance here.

According to all of this, I think that with or without mention changes to be made (I will leave it to the authors to decide) this manuscript definitely and irrevocably should be published.

R: First of all, thank you for your affirmation of my manuscript.

In my manuscript, I consciously reduced the description of the existing methods and increased the description of the experiment process.

In fact, I have not innovation in the existing methods, including PSO algorithm, SVM model and GWR model. What I did, is to find out my research question (to solve the imbalance of results in landslide susceptibility map) and to seek a scientific way to solve it (the GWR-PSO-SVM couple model). In addition, excessive description of existing methods is not conductive for readers (especially for the researchers in the field) to quickly grasp the focus of this paper.

Meanwhile, for the landslide susceptibility map, any step of the experimental process, whether the factor selection or the GWR-based segmentation of the study area proposed in this paper, has a huge impact on the results. In order to ensure the repeatability of the experiment, and other researchers can use their data and my method to get similar conclusions, it is necessary to increase the description of the experiment process.

ROC is a great tool in many fields, especially in medical research. I would be honored to have more in-depth communication with you if you are interested in it.

I hope you are satisfied with this reply.

Reviewer #3:

We first sincerely thank the reviewer for the comments, most of which are very useful for us to improve the paper. Meanwhile, we are sorry that some points of the previous version confused the reviewer. Based on the comments, we have tried our best to improve the manuscript as much as possible. In the following, we will address each comment in detail.

1. In terms of format, this paper is relatively standard. The research methods, contents and results in this paper are all good.

However, there are some little problems in a small amount of languages.

For example,” large landslides were divided into two categories of small & medium and large & detailed scales,”. It’s not clear what are the two categories?

R: First of all, thank you for your affirmation of my manuscript.

As I am from a non-native English-speaking country, I have found a professional retouching company to rewrite the manuscript before I submit it in order to ensure that the language of this paper is closer to the habits of English-speaking countries. But anyway, I apologize for the language problems.

Our modifications are as follows:

In this study, according to the scales and applications of landslide zoning, landslides were divided into two categories: small & medium scales and large & detailed scales, and the results indicated that the guidelines were a “powerful tool for landslide and hazard zoning at different scales”.

I hope you are satisfied with this reply.

---

## [Decision Letter · Decision Letter 1]

18 Feb 2020

A landslide susceptibility map based on spatial scale segmentation: a case study at zigui-badong in the three gorges reservoir area, China

PONE-D-19-27136R1

Dear Dr. Yu,

We are pleased to inform you that your manuscript has been judged scientifically suitable for publication and will be formally accepted for publication once it complies with all outstanding technical requirements.

With kind regards,

Claudionor Ribeiro da Silva

Academic Editor

PLOS ONE

Additional Editor Comments (optional):

Reviewers' comments:

Reviewer's Responses to Questions

**Comments to the Author**

1. If the authors have adequately addressed your comments raised in a previous round of review and you feel that this manuscript is now acceptable for publication, you may indicate that here to bypass the “Comments to the Author” section, enter your conflict of interest statement in the “Confidential to Editor” section, and submit your "Accept" recommendation.

Reviewer #1: (No Response)

Reviewer #2: All comments have been addressed

2. Is the manuscript technically sound, and do the data support the conclusions?

Reviewer #1: Partly

Reviewer #2: Yes

3. Has the statistical analysis been performed appropriately and rigorously? 

Reviewer #1: Yes

Reviewer #2: Yes

4. Have the authors made all data underlying the findings in their manuscript fully available?

Reviewer #1: Yes

Reviewer #2: Yes

5. Is the manuscript presented in an intelligible fashion and written in standard English?

Reviewer #1: No

Reviewer #2: Yes

6. Review Comments to the Author

Reviewer #1: The study presents the results of original research. Experiments, statistics, and other analyses are performed to a technical standard and are described in sufficient detail. Conclusions are presented in an appropriate fashion and are supported by the data. The article be presented in an intelligible fashion and be written in standard English. The article adheres to appropriate reporting guidelines and community standards for data availability.

Reviewer #2: I think that this manuscript would bring the significant contribution to the scientific area. As such it should be available to a broader scientific audience.

7. PLOS authors have the option to publish the peer review history of their article (what does this mean?). If published, this will include your full peer review and any attached files.

Reviewer #1: No

Reviewer #2: Yes: Ivan Grbatinic

---

## [Editor Report · Acceptance letter]

28 Feb 2020

PONE-D-19-27136R1 

A landslide susceptibility map based on spatial scale segmentation: a case study at zigui-badong in the three gorges reservoir area, China 

Dear Dr. Yu:

I am pleased to inform you that your manuscript has been deemed suitable for publication in PLOS ONE. Congratulations! Your manuscript is now with our production department. 

With kind regards,

on behalf of

Dr. Claudionor Ribeiro da Silva 

Academic Editor

PLOS ONE